# Learning to Shape In-distribution Feature Space for Out-of-distribution Detection

**Yonggang Zhang[1], Jie Lu[2], Bo Peng[2], Zhen Fang[2], Yiu-ming Cheung[1]** [*]
[1]Hong Kong Baptist University
[2]Australian Artificial Intelligence Institute, University of Technology Sydney

## Abstract

Out-of-distribution (OOD) detection is critical for deploying machine learning models in the open world. To design scoring functions that discern OOD data from the in-distribution (ID) cases from a pre-trained discriminative model, existing methods tend to make rigorous distributional assumptions either explicitly or implicitly due to the lack of knowledge about the learned feature space in advance. The mismatch between the learned and assumed distributions motivates us to raise a fundamental yet under-explored question: *Is it possible to deterministically model the feature distribution while pre-training a discriminative model?* This paper gives an affirmative answer to this question by presenting a Distributional Representation Learning (DRL) framework for OOD detection. In particular, DRL explicitly enforces the underlying feature space to conform to a pre-defined mixture distribution, together with an online approximation of normalization constants to enable end-to-end training. Furthermore, we formulate DRL into a provably convergent Expectation-Maximization algorithm to avoid trivial solutions and rearrange the sequential sampling to guide the training consistency. Extensive evaluations across mainstream OOD detection benchmarks empirically manifest the superiority of the proposed DRL over its advanced counterparts.

## 1 Introduction

Despite the significant progress in machine learning that has facilitated a broad spectrum of classification tasks [45, 81, 42], models often operate under a *closed-world* scenario, where test data stems from the same distribution as the training data. However, real-world applications often entail scenarios in which deployed models may encounter unseen classes of samples during training, giving rise to what is known as out-of-distribution (OOD) data. These OOD instances can potentially undermine a model's stability and, in certain cases, inflict severe damage on its performance. Accordingly, a reliable discriminative model should not only correctly classify known In-Distribution (ID) samples but also flag any OOD input as "unknown". This directly motivates OOD detection [31, 56, 74], which makes significant differences in ensuring the safety of decision-critical applications, e.g., autonomous driving [26], medical diagnosis [85], and cyber-security [51].

Up to now, a plethora of OOD detection algorithms have been developed recently by leveraging post-hoc analysis on the pre-trained model. The seminal work [22] leverages the maximum softmax probability (MSP), also known as the softmax confidence score, for OOD detection, which is built upon the hypothesis that OOD data should trigger relatively lower softmax confidence than that of ID data. ODIN [35] extends MSP by using temperature scaling and input perturbation to amplify the ID/OOD separability. Approaches of this category, albeit intuitive, are challenged by the observation that deep neural networks are prone to produce over-confident predictions, i.e., abnormally high softmax confidences, even though the inputs are far away from the training data [50]. As a result,

---

[*]Correspondence to Yiu-ming Cheung (ymc@comp.hkbu.edu.hk)

38th Conference on Neural Information Processing Systems (NeurIPS 2024).

advanced methods turn to design alternative OOD scoring functions by resorting to the stored ID patterns in gradients [25], intermediate features [2, 34, 62, 36, 57, 76, 77] and logits [21, 38, 67, 78]. However, most of these methods suffer from the lack of inherent connections and theoretical understandings [48] regarding the detectability of OOD data.

Given that OOD data, by definition, inherently diverges from ID data by means of their data density distributions, some works [37, 48, 54] are motivated to focus on constructing scoring functions that can effectively replicate the behaviour of the ID density function, which opens the door to density-based OOD detection. Despite the empirical success, the power of density-based OOD detection has yet to be fully unleashed even with the helm of feature-shaping [11, 59, 60, 72, 79, 80, 84] and neuron pruning [1, 61]. This is because current methods struggle with making a strong distributional assumption of the underlying feature space either explicitly or implicitly due to the lack of knowledge about the learned feature space in advance. The urgent need to cast off the dilemma that those pre-defined distributions fail to necessarily hold in practice [62] directly promotes the following important yet under-explored question:

> *Is it possible to deterministically shape the ID feature distribution while pre-training a discriminative model?*

**Methodological Contribution.** In this paper, we propose a novel learning framework called Distributional Representation Learning (DRL) that bridges the gap between network pre-training and density-based scoring strategy. At a high level, we explicitly enforce the underlying feature space to conform to a pre-defined distribution, inspired by [41]. In this way, OOD detection can be naturally approached in an assumption-free manner, hence providing stronger flexibility and generality. When implementing our idea, it could always be *non-trivial* to parameterize the ID data distribution since the computation of normalization constants tends to be costly and even intractable [19]. Towards this dilemma, instead of introducing impractical constraints to make the normalization constants input-independent or known, we propose an online approximation of normalization constants to enable end-to-end training.

**Theoretical Contribution.** Importantly, we provide a theoretical framework that formulates DRL as an Expectation-Maximization (EM) algorithm [47]. In particular, by introducing latent variables as ID classes, we obtain the Bayes-optimal inference of the posterior distribution of the latent variables given the observed data in the E-step. In the M-step, we propose to maximize the evidence lower bound (ELBO) of the log conditional likelihood with respect to all parameters so that the optimization process can not only benefit from the classification process but also avoid leading to trivial solutions. We prove that the ELBO is bounded, and the EM algorithm contributes to the convergence of ELBO. Moreover, we rearrange the sequential sampling by constraining half of each mini-batch coinciding with the previous iteration, which deals with the inconsistency issue caused by the integration of the EM algorithm into the batch-based training routine.

**Empirical Contribution.** We extensively evaluate DRL on mainstream OOD detection benchmarks and establish state-of-the-art performance compared with three families of methods: (1) pre-trained with cross-entropy, (2) pre-trained with contrastive learning, (3) pre-trained with cross-entropy and fine-tuned with training-time regularizations. For example, on CIFAR-10, DRL achieves $11.58\%$ FPR95 on average, significantly outperforming PALM [40] by $3.38\%$. Further, for completeness, we extend the evaluation of our method to more strict settings, including (1) large-scale OOD detection, (2) hard OOD detection, and (3) unsupervised OOD detection.

## 2 Preliminary

**Notations.** We write vectors as bold-faced lowercase characters. Considering $K$-way classification as a case study, we use $\mathcal{X}$ and $\mathcal{Y} = \{1, \ldots, K\}$ to indicate the input space and ID label space, respectively. The joint ID distribution, represented as $P_{X_I Y_I}$, is a joint distribution defined over $\mathcal{X} \times \mathcal{Y}$. During testing time, there are some unknown OOD joint distributions $P_{X_O Y_O}$ defined over $\mathcal{X} \times \mathcal{Y}^c$, where $\mathcal{Y}^c$ is the complementary set of $\mathcal{Y}$. We also denote $p(\mathbf{x})$ as the density of the ID marginal distribution $P_{X_I}$. According to [15, 16], OOD detection can be formally defined as follows:

**Problem 1 (OOD Detection)** *Given a labeled ID dataset $\mathcal{D} = \{(\mathbf{x}_1, y_1), ..., (\mathbf{x}_N, y_N)\}$ which is drawn from $P_{X_I Y_I}$ independent and identically distributed, the aim of OOD detection is to learn a*

*predictor $g(\cdot)$ by using $\mathcal{D}$ such that for any test data $\mathbf{x}$: 1) if $\mathbf{x}$ is drawn from $P_{X_{\mathrm{I}}}$, then g can classify $\mathbf{x}$ into correct ID classes, and 2) if $\mathbf{x}$ is drawn from $P_{X_{\mathrm{O}}}$, then g can detect $\mathbf{x}$ as OOD data.*

**OOD Scoring.** Existing methods [25, 62, 34, 67, 37] tend to adopt a post-hoc strategy to detect OOD samples, *i.e.,* given a well-trained discriminative model $f_{\boldsymbol{\theta}} : \mathcal{X} \to \mathbb{R}^d$ using $\mathcal{D}$, and a scoring function $S$, then $\mathbf{x}$ is detected as ID data if and only if $S(\mathbf{x}; f_{\boldsymbol{\theta}}) \geq \lambda$, for some given threshold $\lambda$:

$$g(\mathbf{x}) = \text{ID, if } S(\mathbf{x}; f_{\boldsymbol{\theta}}) \geq \lambda; \text{ otherwise, } g(\mathbf{x}) = \text{OOD.} \tag{1}$$

A natural view for the motivation of the post-hoc strategy is to use a level set for ID density $p(\mathbf{x})$ to discern ID and OOD data. To be specific, the main objective is to construct an efficient scoring function $S$ that can effectively replicate the behaviour of the ID density function $p(\mathbf{x})$ such that $S(\mathbf{x}; f_{\boldsymbol{\theta}}) \propto p(\mathbf{x})$. From this perspective, let $p_{\boldsymbol{\theta}}(\mathbf{x})$ be the ID density function estimated by the discrimination model $f_{\boldsymbol{\theta}}$, we can rewrite Eq. (1) as follow:

$$g(\mathbf{x}) = \text{ID, if } p_{\boldsymbol{\theta}}(\mathbf{x}) \geq \lambda; \text{ otherwise, } g(\mathbf{x}) = \text{OOD.} \tag{2}$$

According to prior works, the design principle for $p_{\boldsymbol{\theta}}(\mathbf{x})$ can be either *logit-based* or *feature-based*.

**Logit-based OOD Methods** derive $p_{\boldsymbol{\theta}}(\mathbf{x})$ by formulating the discriminative model $f_{\boldsymbol{\theta}}$ as an energy-based model [18, 32], where a collection of energy values are turned into a probability density $p_{\boldsymbol{\theta}}(\mathbf{x})$ by implicitly resorting to the Gibbs-Boltzmann distribution, *i.e.,*

$$p_{\boldsymbol{\theta}}(\mathbf{x}) = \frac{\exp\left[-E_{\boldsymbol{\theta}}(\mathbf{z})/\tau\right]}{Z} \propto \exp\left[-E_{\boldsymbol{\theta}}(\mathbf{z})/\tau\right], \quad E_{\boldsymbol{\theta}}(\mathbf{z}) = -\tau \log \sum_{k=1}^{K} \exp\left(r_k/\tau\right), \tag{3}$$

where $Z = \int \exp\left[-E_{\boldsymbol{\theta}}(\mathbf{x})/\tau\right] d\mathbf{x}$ is an *input-independent* normalization constant and $\mathbf{r} = [r_1, ..., r_K] \in \mathbb{R}^K$ denotes the logit vector produced by a classification layer with the intermediate feature $\mathbf{z} \in \mathbb{R}^d$ as the input. $\tau > 0$ denotes a temperature hyperparameter and, when $\tau \to 0$, the negative energy score [37] $-E_{\boldsymbol{\theta}}(\cdot)$ will degenerate into the MaxLogit score [21, 78], namely,

$$\lim_{\tau \to 0} -E_{\boldsymbol{\theta}}(\mathbf{z}) = \lim_{\tau \to 0} \tau \log \sum_{k=1}^{K} \exp\left(r_k/\tau\right) = \max_{k \in \mathcal{Y}} r_k.$$

**Feature-based OOD Methods** derive $p_{\boldsymbol{\theta}}(\mathbf{x})$ by explicitly assuming the intermediate feature space $\mathcal{Z}$ learned by the discriminative model $f_{\boldsymbol{\theta}}$ to follow a pre-defined distribution. Therein, a representative example [48] is a Gaussian mixture distribution under a uniform ID-class prior (*i.e.,* $p_{\boldsymbol{\theta}}(k) = 1/K$). Formally, let $\mathcal{N}(\boldsymbol{\mu}_k, \tau\Sigma)$ denotes the $k$-th ID class-conditional Gaussian distribution with a tied covariance matrix $\tau\Sigma$, then we have

$$p_{\boldsymbol{\theta}}(\mathbf{x}) = \frac{1}{K} \sum_{k=1}^{K} \frac{\exp\left[-\frac{1}{2}(\mathbf{z} - \boldsymbol{\mu}_k)^\top (\tau\Sigma)^{-1}(\mathbf{z} - \boldsymbol{\mu}_k)\right]}{\sqrt{(2\pi)^d |t\Sigma|}} \propto \sum_{k=1}^{K} \exp\left[-\frac{1}{2}(\mathbf{z} - \boldsymbol{\mu}_k)^\top (\tau\Sigma)^{-1}(\mathbf{z} - \boldsymbol{\mu}_k)\right]. \tag{4}$$

Clearly, Eq. (4) is equivalent to the GEM score [34] when $t = 1$. Reducing $\tau$ gradually reinforces the intra-class intensity and $\tau \to 0$ degenerates Eq. (4) into the maximum Mahalanobis distance [34] (up to a constant), *i.e.,*

$$\lim_{\tau \to 0} \tau \log p_{\boldsymbol{\theta}}(\mathbf{x}) = \lim_{\tau \to 0} \tau \log \frac{1}{K} \sum_{k=1}^{K} \frac{\exp\left[-\frac{1}{2}(\mathbf{z} - \boldsymbol{\mu}_k)^\top (\tau\Sigma)^{-1}(\mathbf{z} - \boldsymbol{\mu}_k)\right]}{\sqrt{(2\pi)^d |\tau\Sigma|}}$$

$$= \lim_{\tau \to 0} \tau \log \sum_{k=1}^{K} \exp\left[-\frac{1}{2}(\mathbf{z} - \boldsymbol{\mu}_k)^\top \Sigma^{-1}(\mathbf{z} - \boldsymbol{\mu}_k)/\tau\right] - \lim_{\tau \to 0} \tau \log K \sqrt{(2\pi)^d |\tau\Sigma|}$$

$$= \max_{k \in \mathcal{Y}} -(\mathbf{z} - \boldsymbol{\mu}_k)^\top \Sigma^{-1}(\mathbf{z} - \boldsymbol{\mu}_k) - \underbrace{\lim_{\tau \to 0} \tau \log K \sqrt{(2\pi)^d |\tau\Sigma|}}_{\text{constant } \forall \mathbf{x} \in \mathcal{X}}.$$

## 3 Methodology

### 3.1 Motivation

We hereby present the motivation behind our work. In light of the aforementioned discussion, it can be found that existing OOD detection methods tend to make distributional assumptions on $p_{\boldsymbol{\theta}}(\mathbf{x})$

either explicitly or implicitly with the aim of density estimation due to the lacking true knowledge of $p_{\boldsymbol{\theta}}(\mathbf{x})$. However, these distributional assumptions would not be held in many practical scenarios, leading to the mismatch between distributions of the learned and assumed features.

To illustrate this for the logit-based methods, we forge a mathematical connection between the classification objective and ID density function $p_{\boldsymbol{\theta}}(\mathbf{x})$:

$$\log \frac{\exp\left(r_y/\tau\right)}{\sum_{k=1}^{K} \exp\left(r_k/\tau\right)} = r_y/\tau - \log \sum_{k=1}^{K} \exp\left(r_k/\tau\right) = r_y/\tau - \log Z p_{\boldsymbol{\theta}}(\mathbf{x}), \tag{5}$$

where $Z$ is the constant in Eq. 3. Maximizing the classification objective in Eq. 5 is to maximize the logit $r_k$ while minimizing the estimated density function $p_{\boldsymbol{\theta}}(\mathbf{x})$. This implies that training a discriminative model drives the ID marginal distribution away from the Gibbs-Boltzmann case. This cannot support the idea that the ID marginal distribution can necessarily align with the Gaussian distribution in the feature-based ones. Furthermore, it has been argued by [17] that the learned latent features fail the Henze-Zirkler multivariate normality test [24], which challenges the rationality of Gaussian-based distributional assumptions. We notice the possibility of non-parametric density estimation based on nearest-neighbor distance [17, 62] in the latent feature space. Albeit assumption-free, this practice hampers the scalability as it takes $\mathcal{O}(Nd)$ memory to store the latent feature of ID training samples and $\mathcal{O}(Nd)$ computation for neighborhood discovery. Most recently, [54] empirically shows that parametric density estimation can significantly benefit OOD detection more than the non-parametric counterpart if more reasonable distributional assumptions are introduced.

However, the bottleneck of OOD detection still lies in the consistency between the assumed distribution and the unknown ground truth. In view of this, this paper proposes to deterministically shape the learned feature distribution while training a discriminative model so that we can relax the widely adopted distributional assumptions.

## 3.2 Data distribution modeling

To preserve inter-class structures, this work, following common practice [13, 46], explicitly models the ID marginal distribution $P_{X_{\mathrm{I}}}$ as a weighted aggregation of ID class-conditioned distributions, *i.e.,*

$$\log p_{\boldsymbol{\theta}}(\mathbf{x}) = \log \sum_{k=1}^{K} p_{\boldsymbol{\theta}}(\mathbf{x}|k) \cdot p_{\boldsymbol{\theta}}(k). \tag{6}$$

Consistent with probabilistic theory, $p_{\boldsymbol{\theta}}(\mathbf{x}|k)$ can be formulated into the following general form:

$$p_{\boldsymbol{\theta}}(\mathbf{x}|k) = \frac{h(\mathbf{z}, k)}{\Phi(k)}, \quad \Phi(k) = \int_{\mathbf{z} \in \mathcal{Z}} h(\mathbf{z}, k)\, d\mathbf{z}, \tag{7}$$

where $h(\mathbf{z}, k)$ represents a non-negative density function defined over the lower-dimensional latent feature space in which high-frequency and imperceptible details are abstracted away [55]. Note that our method is generic to the choice of $h$ while this paper, inspired by [69], focuses on an exemplar based on the Euclidean distance, *i.e.,* $h(\mathbf{z}, k) = \exp\left(-\|\mathbf{z} - \boldsymbol{\mu}_k\|_2^2 / 2\tau\right)$, that is closely connected to universally optimal point configurations [4, 8]. Motivated by [13, 46], we further introduce $\ell_2$ normalization over the latent feature space to keep $\|\mathbf{z}\|_2 = 1$. Without loss of generalization, let $\hat{\boldsymbol{\mu}}_{kj} = \boldsymbol{\mu}_k / \|\boldsymbol{\mu}_k\|_2$ and $\epsilon_k = \|\boldsymbol{\mu}_k\|_2 / \tau$, we then have:

$$p_{\boldsymbol{\theta}}(\mathbf{x}|k) = \frac{\exp\left[-(1 - 2 \cdot \boldsymbol{\mu}_k^\top \mathbf{z} + \|\boldsymbol{\mu}_k\|_2^2)/2\tau\right]}{\int_{\mathbf{z}' \in \mathcal{Z}} \exp\left[-(1 - 2 \cdot \boldsymbol{\mu}_k^\top \mathbf{z}' + \|\boldsymbol{\mu}_k\|_2^2)/2\tau\right] d\mathbf{z}'} = \frac{\exp(\epsilon_k \hat{\boldsymbol{\mu}}_k^\top \mathbf{z})}{\pi(\hat{\boldsymbol{\mu}}_k, \epsilon_k)}, \tag{8}$$

where $\pi(\hat{\boldsymbol{\mu}}_k, \epsilon_k) = \int_{\mathbf{z} \in \mathcal{Z}} \exp\left(\epsilon_k \hat{\boldsymbol{\mu}}_k^\top \mathbf{z}\right) d\mathbf{z}$.

## 3.3 Online approximation of normalization constant $\pi(\hat{\boldsymbol{\mu}}_k, \epsilon_k)$

Regarding the computation of $p_{\boldsymbol{\theta}}(\mathbf{x}|k)$ in Eq. (8), it is imperative to accurately calculate the normalization constant $\pi(\hat{\boldsymbol{\mu}}_k, \epsilon_k)$ that seems to be intractable due to the integral over the latent feature space. When connecting Eq. (8) with the well-known von Mises-Fisher (vMF) distribution, one

can easily check that the normalization constant $\pi(\hat{\boldsymbol{\mu}}_k, \epsilon_k) = \epsilon_k^{d/2-1} / \left[ (2\pi)^{d/2} I_{d/2-1}(\epsilon_k) \right]$ where $I_p(\cdot)$ denotes the modified Bessel function of the first kind and order $p$. Unfortunately, $I_p(\cdot)$ is a complicated function. To cast off this dilemma, the most straightforward idea, as in [13, 40, 46], involves fixing the $\ell_2$ norm of $\boldsymbol{\mu}_k$ to be a constant value, $e.g.,$ $\|\boldsymbol{\mu}_k\|_2 = 1, \forall k$. This will result in $\epsilon_k = 1/\tau, \forall k$ and therefore allows one to bypass the computation of the normalization constant $\pi(\hat{\boldsymbol{\mu}}_k, \epsilon_k)$ as $\pi(\hat{\boldsymbol{\mu}}_k, \epsilon_k)$ turns to be a class-independent constant term during optimization. Despite the simplicity, it is crucial to acknowledge that this practice implicitly implies the rigorous assumption that all classes ought to have a similar level of concentration, and would likely reduce the flexibility of the mixture model in Eq. (6). Indeed, this inflexibility in the latent feature space tends to deteriorate the learning of the feature mapping function. In this paper, we proceed from a different perspective and propose to use an online approximation thereof. To be precise, with a sufficiently large $p$, $I_p(\cdot)$ can be approximated using the following uniform expansion [39], $i.e.,$

$$I_p(pq) \sim \frac{\exp(p\nu)}{(2\pi p)^{1/2}(1+q^2)^{1/4}} \sum_{j=0}^{\infty} \frac{U_j(s)}{p^j}, \quad \nu = (1+q^2)^{1/2} + \log \frac{q}{1+(1+q^2)^{1/2}} \quad (9)$$

where $\sim$ denotes a Poincaré asymptotic expansion with regard to $p$, $s = (1+q^2)^{1/2}$ and polynomials $U_j(s)$. Empirically, we find that $p = 127$, which corresponds to the latent feature dimension $d = 256$, and evaluating only the first term in the sum, $i.e.,$ $U_0(s) = 1$, are sufficiently enough to contribute to a satisfactory approximation while keeping simplicity. Please refer to Appendix A for more details.

### 3.4 DRL as Expectation-Maximization

In practice, it is hard to directly optimize the log-likelihood function [3, 64, 68] due to the existence of trivial solutions in which the latent feature space either scales up towards infinity or collapses to the origin point. To avoid collapse while making the classification process maximally beneficial to distribution modeling, let us start from the following evidence lower bound (ELBO) of $\log p_{\boldsymbol{\theta}}(\mathbf{x})$:

$$\log p_{\boldsymbol{\theta}}(\mathbf{x}) = \sum_{k=1}^{K} q(k|\mathbf{x}) \log p_{\boldsymbol{\theta}}(\mathbf{x}|k) - \mathrm{KL}\left[q(k|\mathbf{x}) \,\|\, p_{\boldsymbol{\theta}}(k)\right] + \mathrm{KL}\left[q(k|\mathbf{x}) \,\|\, p_{\boldsymbol{\theta}}(k|\mathbf{x})\right]$$
$$\geq \sum_{k=1}^{K} q(k|\mathbf{x}) \log p_{\boldsymbol{\theta}}(\mathbf{x}|k) - \mathrm{KL}\left[q(k|\mathbf{x}) \,\|\, p_{\boldsymbol{\theta}}(k)\right] = \mathrm{ELBO}(q, \mathbf{x}; \boldsymbol{\theta}), \quad (10)$$

where $\mathrm{KL}\left[\cdot\right]$ denotes the Kullback–Leibler (KL) divergence. We derive this ELBO in Appendix B. To make the inequality hold with equality so that the ELBO reaches its maximum value $\log p(\mathbf{x})$, we require $\mathrm{KL}\left[q(k|\mathbf{x}) \,\|\, p_{\boldsymbol{\theta}}(k|\mathbf{x})\right] = 0$. By replacing $q(k|\mathbf{x})$ with $p_{\boldsymbol{\theta}}(k|\mathbf{x})$, we can formulate the maximization of the ELBO into the expectation–maximization (EM) framework.

**E-step.** With the fixed $\boldsymbol{\theta}_t$ at the iteration $t$, this step aims to estimate $q_{t+1}(k|\mathbf{x})$ to make $q_{t+1}(k|\mathbf{x}) = p_{\boldsymbol{\theta}_t}(k|\mathbf{x})$ so that we can have $\mathrm{ELBO}(q_{t+1}, \mathbf{x}; \boldsymbol{\theta}_t) = p_{\boldsymbol{\theta}_t}(\mathbf{x})$. Here, we estimate $p_{\boldsymbol{\theta}_t}(k|\mathbf{x})$ with the soft labels produced by the discrimination model at the iteration $t$, $i.e.,$ $f$ parameterized by $\boldsymbol{\theta}_t$. To this end, by applying Bayes' theorem to $p_{\boldsymbol{\theta}_t}(\mathbf{x})$, we approximate $q_{t+1}(k|\mathbf{x})$ as:

$$q_{t+1}(k|\mathbf{x}) = \frac{p_{\boldsymbol{\theta}}(\mathbf{x}|k) p_{\boldsymbol{\theta}}(k)}{\sum_{c=1}^{K} p_{\boldsymbol{\theta}}(\mathbf{x}|c) p_{\boldsymbol{\theta}}(c)} = \frac{\exp(\epsilon_y \hat{\boldsymbol{\mu}}_y^\top \mathbf{z}) / \pi(\hat{\boldsymbol{\mu}}_y, \epsilon_y)}{\sum_{k=1}^{K} \exp(\epsilon_k \hat{\boldsymbol{\mu}}_k^\top \mathbf{z}) / \pi(\hat{\boldsymbol{\mu}}_k, \epsilon_k)} \quad (11)$$

**M-step.** With the sub-optimal $q_{t+1}(k|\mathbf{x}) = p_{\boldsymbol{\theta}_t}(k|\mathbf{x})$ after E-step, we turn to maximize the ELBO. Given that $p_{\boldsymbol{\theta}}(k) = 1/K$ and $q_{t+1}(k|\mathbf{x})$ is fixed during optimization, the KL term in $\mathrm{ELBO}(q, \mathbf{x}; \boldsymbol{\theta})$ can be reduced to a constant form $-\mathrm{KL}\left[q(k|\mathbf{x}) \,\|\, p_{\boldsymbol{\theta}}(k)\right] = \log K + H\left[q(k|\mathbf{x})\right]$, which results in:

$$\boldsymbol{\theta}_{t+1} = \arg\max_{\boldsymbol{\theta}_t} \mathbb{E}_{(\mathbf{x}, y) \in \mathcal{D}} \left[ \sum_{k=1}^{K} q(k|\mathbf{x}) \log p_{\boldsymbol{\theta}_t}(\mathbf{x}|k) \right]. \quad (12)$$

**Convergence.** At the E-step of the iteration $t$, we estimate $q_{t+1}(k|\mathbf{x})$ to ensure $\mathrm{ELBO}(q_{t+1}, \mathbf{x}; \boldsymbol{\theta}_t) = \log p_{\boldsymbol{\theta}_t}(k|\mathbf{x})$. At the M-step after the E-step, we have obtained $\boldsymbol{\theta}_{t+1}$ with a fixed $q_{t+1}(k|\mathbf{x})$ such that $\mathrm{ELBO}(q_{t+1}, \mathbf{x}; \boldsymbol{\theta}_{t+1}) \geq \mathrm{ELBO}(q_{t+1}, \mathbf{x}; \boldsymbol{\theta}_t)$. Accordingly, we arrive at the following sequence:

$$\log p_{\boldsymbol{\theta}_{t+1}}(\mathbf{x}) \geq \mathrm{ELBO}(q_{t+1}, \mathbf{x}; \boldsymbol{\theta}_{t+1}) \geq \mathrm{ELBO}(q_{t+1}, \mathbf{x}; \boldsymbol{\theta}_t) = \log p_{\boldsymbol{\theta}_t}(\mathbf{x}). \quad (13)$$

Since $\log p_{\boldsymbol{\theta}_{t+1}}(k|\mathbf{x}) \geq \log p_{\boldsymbol{\theta}_t}(k|\mathbf{x})$, one can guarantee that $\text{ELBO}(q, \mathbf{x}; \boldsymbol{\theta})$ is upper-bounded and can converge to a certain value with the proposed EM framework.

**Implementation.** When integrating the EM framework into the batch-based training routine, we note that the optimization of $\text{ELBO}(q, \mathbf{x}; \boldsymbol{\theta})$ would suffer from the inconsistency issue. This is because the prediction $p_{\boldsymbol{\theta}}(k|\mathbf{x})$ only gets updated when $\mathbf{x}$ is fed as the input. In other words, the prediction $p_{\boldsymbol{\theta}}(k|\mathbf{x})$ is only updated once per training epoch. However, the discriminative model $f_{\boldsymbol{\theta}}$ keeps updated throughout mini-batches in the training epoch. To address this problem, we rearrange the sequential sampling by constraining half of each mini-batch coinciding with the previous mini-batch. Meanwhile, the rest of the half will coincide with the next mini-batch. In this way, half of the current mini-batch can obtain on-the-fly soft labels generated by the discriminative model $f_{\boldsymbol{\theta}}$ in the previous mini-batch. For a clear notation, let $\mathcal{B}_m = \{\mathcal{B}_m^{\text{pre}}, \mathcal{B}_m^{\text{next}}\}$ be the $m$-th mini-batch sampled from $\mathcal{D}$ in the current training epoch, we then have $\mathcal{B}_{m-1}^{\text{next}} = \mathcal{B}_m^{\text{pre}}$ and $\mathcal{B}_1^{\text{pre}} = \emptyset$.

### 3.5 Overall training objective

Our method enables end-to-end training, where the overall training objective $\mathcal{R}(\cdot; \boldsymbol{\theta})$ is to maximize a linear combination of the classification objective $\mathcal{R}_{\text{cls}}(\cdot, \cdot; \boldsymbol{\theta})$ and the ELBO in Eq. (10), *i.e.*,

$$\mathcal{R}(\mathcal{B}_m; \boldsymbol{\theta}) = \mathbb{E}_{(\mathbf{x},y) \in \mathcal{B}_m} \mathcal{R}_{\text{cls}}(\mathbf{x}, y; \boldsymbol{\theta}) + \beta \mathbb{E}_{(\mathbf{x},y) \in \mathcal{B}_m^{\text{pre}}} \text{ELBO}(q, \mathbf{x}; \boldsymbol{\theta}), \tag{14}$$

where the hyperparameter $\beta > 0$ modulates the relative importance of two losses. Based on our proposed distribution modeling in Eq. (6), the classification for a sample $\mathbf{x}$ can take place with a Bayes-based rule instead of a parametric softmax layer. As such, the the classification objective $\mathcal{R}_{\text{cls}}(\mathbf{x}, y; \boldsymbol{\theta})$ naturally turns to be:

$$\mathcal{R}_{\text{cls}}(\mathbf{x}, y; \boldsymbol{\theta}) = \log \frac{\exp(\epsilon_y \hat{\boldsymbol{\mu}}_y^\top \mathbf{z}) / \pi(\hat{\boldsymbol{\mu}}_y, \epsilon_y)}{\sum_{k=1}^K \exp(\epsilon_k \hat{\boldsymbol{\mu}}_k^\top \mathbf{z}) / \pi(\hat{\boldsymbol{\mu}}_k, \epsilon_k)}. \tag{15}$$

**Remark.** While recent works [13, 40, 46] come with a Bayes-based classification objective as well by modeling the latent feature space as a mixture of vMF distributions. However, we emphasize that optimizing the classification objective alone does not necessarily drive the extracted training features towards the pre-defined distribution. For example, a latent feature $\mathbf{z}$ can be far away from the corresponding class prototype $\boldsymbol{\mu}_y$ while still being correctly classified as long as $\mathbf{z}$ is relatively closer to $\boldsymbol{\mu}_y$ than to other class prototypes. We, accordingly, mitigate this problem with the ELBO in Eq. (10) to explicitly measure the extent to which a training sample fits the assumed distribution. Besides, these methods are limited by assuming the concentration parameter $\epsilon_k$ to be class-uniform for a simplified computation of Eq. (15) even though [13] considers optimizing $\epsilon_k$ along with the discriminative model. By contrast, our method learns $\epsilon_k$ directly through the prototype magnitudes while proposing an online approximation of the normalization constant $\pi(\hat{\boldsymbol{\mu}}_k, \epsilon_k)$ to enable end-to-end training.

## 4 Related work

**Out-of-Distribution Detection** has attracted a surge of interest in recent years, which is motivated by the empirical observation [50] that neural networks tend to be over-confident in OOD data [52]. One line of work performs OOD detection by devising post-hoc scoring functions, including confidence-based methods [21, 38, 78], energy-based methods [37, 67], distance-based approaches [2, 34, 62, 36, 57, 76, 77], gradient-based approaches [25], and Bayesian approaches [29, 44]. Another line of work addresses OOD detection by fine-tuning a pre-trained discrimination model with training-time regularizations that help the model learn ID/OOD discrepancy following the guideline of outlier exposure [23] or negative prompts [53]. For instance, the discriminative model is regularized to produce lower confidence [33, 43], smaller feature magnitudes [37] or higher energy [10] for outlier points. More recently, some works have considered a practical scenario where the auxiliary outliers can be arbitrarily different from the real OOD data, therefore, distributionally augmenting the observed OOD data. Besides, the given OOD samples tend to include unlabelled ID counterparts [27]. In view of this, WOOD [27] formulates learning with noisy OOD samples as a constrained optimization problem while SAL [12] separates candidate outliers from the unlabeled data and then trains a binary classifier using the candidate outliers and the labeled ID data. Most regularization methods, unfortunately, assume the availability of auxiliary OOD data, not to mention resource-intensive re-training processes, while our method maintains the same training scheme as standard cross-entropy loss without requiring additional OOD data in training.

Table 1: OOD detection results on the CIFAR-10 benchmark with ResNet-18. ↑ indicates larger values are better and vice versa. The best results in the last two columns are shown in bold.

| Method | SVHN | | Places365 | | LSUN | | iSUN | | Texture | | Average | |
|---|---|---|---|---|---|---|---|---|---|---|---|---|
| | FPR95↓ | AUROC↑ | FPR95↓ | AUROC↑ | FPR95↓ | AUROC↑ | FPR95↓ | AUROC↑ | FPR95↓ | AUROC↑ | FPR95↓ | AUROC↑ |
| MSP | 59.66 | 91.25 | 62.46 | 88.64 | 51.93 | 92.73 | 54.57 | 92.12 | 66.45 | 88.50 | 59.01 | 90.65 |
| ODIN | 20.93 | 95.55 | 63.04 | 86.57 | 31.92 | 94.82 | 33.17 | 94.65 | 56.40 | 86.21 | 41.09 | 91.56 |
| Energy | 54.41 | 91.22 | 37.22 | 92.70 | 10.19 | 98.05 | 27.52 | 95.59 | 55.23 | 89.37 | 36.91 | 93.39 |
| ReAct | 48.16 | 92.32 | 37.25 | 93.13 | 18.09 | 96.91 | 20.35 | 95.59 | 96.51 | 47.41 | 34.25 | 94.09 |
| ASH | 28.94 | 94.84 | 27.29 | 91.31 | 9.06 | 98.34 | 21.61 | 95.95 | 35.02 | 93.63 | 27.29 | 94.81 |
| Maha | 9.24 | 97.80 | 83.50 | 69.56 | 67.73 | 73.61 | 6.02 | 98.63 | 23.21 | 92.91 | 37.94 | 86.50 |
| KNN | 27.97 | 95.48 | 47.84 | 89.93 | 18.50 | 96.84 | 24.68 | 95.52 | 26.74 | 94.96 | 29.15 | 94.55 |
| CONJ | 18.71 | 96.48 | 53.44 | 89.18 | 22.20 | 95.95 | 22.64 | 95.87 | 25.80 | 95.11 | 28.56 | 94.52 |
| Vim | 24.95 | 95.36 | 63.04 | 86.57 | 7.26 | 98.53 | 33.17 | 94.65 | 56.40 | 86.21 | 36.96 | 92.26 |
| VOS | 15.69 | 96.37 | 37.95 | 91.78 | 27.64 | 93.82 | 30.42 | 94.87 | 32.68 | 93.68 | 28.88 | 94.10 |
| CSI | 37.38 | 94.69 | 38.31 | 93.04 | 10.63 | 97.93 | 10.36 | 98.01 | 28.85 | 94.87 | 25.11 | 95.71 |
| SSD+ | 2.47 | 99.51 | 22.05 | 95.57 | 10.56 | 97.83 | 28.44 | 95.67 | 9.27 | 98.35 | 14.56 | 97.38 |
| KNN+ | 2.70 | 99.61 | 23.05 | 94.88 | 7.89 | 98.01 | 24.56 | 96.21 | 10.11 | 97.43 | 13.66 | 97.22 |
| CIDER | 2.89 | 99.72 | 23.05 | 94.09 | 5.45 | 99.01 | 20.21 | 96.64 | 12.33 | 96.85 | 12.95 | 97.26 |
| PALM | 0.34 | 99.91 | 28.81 | 94.80 | 1.11 | 99.65 | 34.07 | 95.17 | 10.48 | 98.29 | 14.96 | 97.57 |
| DRL | 7.91 | 98.82 | 19.17 | 95.65 | 12.87 | 99.09 | 11.92 | 98.12 | 4.92 | 97.48 | **11.58** | **97.83** |

**Representation Learning for OOD Detection** is an emerging topic to enhance the performance of post-hoc distance-based OOD detectors. In particular, CSI [63] investigates the type of data augmentations that are particularly beneficial for OOD detection while other works [58, 70] verify the effectiveness of applying the off-the-shelf multi-view contrastive losses such as SimCLR [5] and SupCon [28] for OOD detection. CIDER [46] proposes a prototypical contrastive learning framework for OOD detection by promoting stronger ID-OOD separability than SupCon loss, where a regularization strategy is to ensure that all samples are compactly located around their corresponding class prototype. PALM [40] extends CIDER by introducing a mixture of prototypes to represent each class and performing prototype-level contrastive learning to enhance intra-class compactness and inter-class discrimination. This paper significantly differs from PALM and CIDER in the following two aspects: 1) we avoid normalizing the class prototypes to protect the generalization ability of the data distributional modeling, and 2) we learn a feature space that best describes the pre-defined distribution rather than features that preserve inter- and intra- class structures.

# 5 Experiments

**Software and Hardware**. We perform all experiments on an NVIDIA A100 GPU using Pytorch.

**Baseline Methods**. We compare DRL with representative methods, including MSP [22], ODIN [35], Energy [37], ReAct [37], ASH [11], Mahalanobis (Maha) [34], KNN [62], CONJ [54], Vim [66], VOS [14], CSI [63], SSD+ [58], KNN [37], CIDER [46], and PALM [40]. It is worth noting that we have adopted the recommended configurations proposed by prior works, while concurrently standardizing the backbone architecture to ensure equitable comparisons.

**Evaluation Metrics**. The detection performance is evaluated via three widely used metrics: 1) the false positive rate of OOD data is measured when the true positive rate of ID data reaches 95% (FPR95); 2) the area under the receiver operating characteristic curve (AUROC) is computed to quantify the probability of the ID case receiving a higher score than the OOD case. The reported results of DRL are averaged over 5 independent runs.

## 5.1 Main results on CIFAR benchmarks

Following the setup in [46, 40], we consider CIFAR-10 [30] and CIFAR-100 [30] as ID datasets and train ResNet-18 [20] and ResNet-34 [20] on them respectively. We train the model using stochastic gradient descent with momentum 0.9, and weight decay $10^{-4}$ for 500 epochs. The initial learning rate is 0.5 with cosine scheduling and the batch size is 512. There are six datasets for OOD detection with regard to CIFAR benchmarks: SVHN [49], LSUN [75], iSUN [73], Places [82], and Textures [6]. At inference time, all images are of size 32×32. Tables 1 and 2 presented the performance of different methods, where our method significantly outperforms existing methods. Specifically, compared with the mostly advanced PALM, DRL reveals 3.38% average improvements w.r.t. FPR95 on CIFAR-10.

Table 2: OOD detection results on the CIFAR-100 benchmark with ResNet-34. ↑ indicates larger values are better and vice versa. The best results in the last two columns are shown in bold.

| Method | SVHN | | Places365 | | LSUN | | iSUN | | Texture | | Average | |
|---|---|---|---|---|---|---|---|---|---|---|---|---|
| | FPR95↓ | AUROC↑ | FPR95↓ | AUROC↑ | FPR95↓ | AUROC↑ | FPR95↓ | AUROC↑ | FPR95↓ | AUROC↑ | FPR95↓ | AUROC↑ |
| MSP | 78.89 | 79.80 | 84.38 | 74.21 | 83.47 | 75.28 | 84.61 | 74.51 | 86.51 | 72.53 | 83.12 | 75.27 |
| ODIN | 70.16 | 84.88 | 82.16 | 75.19 | 76.36 | 80.10 | 79.54 | 79.16 | 85.28 | 75.23 | 78.70 | 79.11 |
| Energy | 66.91 | 85.25 | 81.41 | 76.37 | 59.77 | 86.69 | 66.52 | 84.49 | 79.01 | 79.96 | 70.72 | 82.55 |
| ReAct | 50.93 | 88.75 | 83.55 | 73.10 | 64.02 | 80.31 | 81.80 | 79.99 | 64.40 | 81.95 | 68.94 | 80.82 |
| ASH | 52.96 | 90.19 | 72.62 | 76.38 | 75.18 | 76.52 | 55.55 | 87.86 | 56.17 | 86.75 | 62.50 | 83.53 |
| Maha | 87.09 | 80.62 | 84.63 | 73.89 | 84.15 | 79.43 | 83.18 | 78.83 | 61.72 | 84.87 | 80.15 | 79.53 |
| KNN | 46.25 | 90.39 | 82.08 | 75.44 | 60.85 | 85.61 | 71.56 | 86.28 | 62.39 | 83.95 | 64.63 | 84.33 |
| CONJ | 46.19 | 90.44 | 80.81 | 75.83 | 60.45 | 85.90 | 64.62 | 87.77 | 62.13 | 83.77 | 62.84 | 84.74 |
| Vim | 73.42 | 84.62 | 85.34 | 69.34 | 86.96 | 69.74 | 85.35 | 73.16 | 74.56 | 76.23 | 81.13 | 74.62 |
| VOS | 43.24 | 82.80 | 76.85 | 78.63 | 73.61 | 84.69 | 69.65 | 86.32 | 57.57 | 87.31 | 64.18 | 83.95 |
| CSI | 44.53 | 92.65 | 79.08 | 76.27 | 75.58 | 83.78 | 76.62 | 84.98 | 61.61 | 86.47 | 67.48 | 84.83 |
| SSD+ | 31.19 | 94.19 | 77.74 | 79.90 | 79.39 | 85.18 | 80.85 | 84.08 | 66.63 | 86.18 | 67.16 | 85.91 |
| KNN+ | 39.23 | 92.78 | 80.74 | 77.58 | 48.99 | 89.30 | 74.99 | 82.69 | 57.15 | 88.35 | 60.22 | 86.14 |
| CIDER | 23.09 | 95.16 | 79.63 | 73.43 | 16.16 | 96.33 | 71.68 | 82.98 | 43.87 | 90.42 | 46.89 | 87.67 |
| PALM | 3.29 | 99.23 | 64.66 | 84.72 | 9.86 | 98.01 | 28.71 | 94.64 | 33.56 | 92.49 | **28.02** | **93.82** |
| DRL | 20.15 | 94.07 | 76.64 | 77.55 | 16.97 | 94.63 | 32.57 | 92.33 | 31.97 | 92.09 | 35.66 | 90.13 |

## 5.2 Extensions

**Hard OOD Detection.** We consider hard OOD scenarios, in which the OOD data are semantically similar to those of the ID cases. With the CIFAR-100 as the ID dataset for training ResNet-34. we evaluate our method on 4 hard OOD datasets, namely, LSUN-Fix [75], ImageNet-Fix [9], ImageNet-Resize [9], and CIFAR-10. We select a set of strong baselines that are competent in hard OOD detection with results in Table 3. It can be found that our method can beat the state-of-the-art across the considered datasets, even for the challenging CIFAR-100 versus CIFAR-10 setting.

Table 3: Evaluation on hard OOD detection tasks. ↑ indicates larger values are better and vice versa. The best result in each column is shown in bold.

| Method | LSUN-Fix | | ImageNet-Fix | | ImageNet-Resize | | CIFAR-10 | | Average | |
|---|---|---|---|---|---|---|---|---|---|---|
| | FPR95↓ | AUROC↑ | FPR95↓ | AUROC↑ | FPR95↓ | AUROC↑ | FPR95↓ | AUROC↑ | FPR95↓ | AUROC↑ |
| SSD+ | 83.36 | 76.63 | 76.73 | 79.78 | 83.67 | 81.09 | 85.16 | 73.70 | 82.23 | 77.80 |
| KNN+ | 84.96 | 75.37 | 75.52 | 79.95 | 68.49 | 84.91 | 84.12 | 75.91 | 78.27 | 79.04 |
| CIDER | 90.94 | 70.31 | 78.83 | 77.53 | 56.89 | 87.62 | 84.87 | 73.30 | 77.88 | 77.19 |
| PALM | 77.15 | 77.24 | 66.19 | 82.51 | 27.02 | 95.03 | 87.25 | 72.28 | 64.40 | 81.76 |
| DRL | 68.64 | 78.75 | 59.92 | 87.19 | 40.85 | 91.11 | 74.12 | 80.93 | **60.88** | **84.50** |

**Unsupervised OOD Detection.** To verify the reliance of our method on the availability of ground-truth labels, we consider unsupervised OOD scenarios. Following [40], we take unlabelled CIFAR-100 as the ID dataset to train ResNet-34 from scratch. Due to the lack of ground-truth labels, we resort to maintaining a momentum teacher, inspired by DINO [83], to produce soft pseudo-labels as a surrogate. For a fair comparison, we use the same data augmentation techniques with [40].

Table 4: Evaluation on unsupervised OOD detection tasks. ↑ indicates larger values are better and vice versa. The best results in the last two columns are shown in bold.

| Method | SVHN | | Places365 | | LSUN | | iSUN | | Texture | | Average | |
|---|---|---|---|---|---|---|---|---|---|---|---|---|
| | FPR95↓ | AUROC↑ | FPR95↓ | AUROC↑ | FPR95↓ | AUROC↑ | FPR95↓ | AUROC↑ | FPR95↓ | AUROC↑ | FPR95↓ | AUROC↑ |
| KNN | 61.21 | 84.92 | 81.46 | 72.97 | 69.65 | 77.77 | 93.35 | 70.39 | 78.49 | 76.75 | 74.83 | 76.56 |
| SSD | 60.13 | 86.40 | 79.05 | 73.68 | 61.94 | 84.47 | 84.37 | 75.58 | 71.91 | 83.35 | 71.48 | 80.70 |
| CSI | 14.47 | 97.14 | 86.23 | 66.93 | 34.12 | 94.21 | 87.79 | 80.15 | 80.15 | 92.13 | 53.55 | 86.11 |
| PALM | 13.86 | 97.53 | 85.63 | 69.46 | 21.28 | 95.95 | 53.43 | 89.06 | 42.62 | 88.33 | 43.37 | 88.07 |
| DRL | 25.11 | 94.50 | 79.06 | 74.10 | 36.88 | 90.88 | 30.66 | 93.96 | 31.59 | 91.58 | **40.66** | **89.00** |

**Large-scale OOD Detection.** We conduct experiments on the ImageNet benchmark, demonstrating the scalability of our method. Specifically, we inherit the setup from [40, 46], where the ID dataset is ImageNet-100 [9], and OOD datasets include iNaturalist [65], SUN [71], Places365 [83], and

Textures [7]. Following the setting in the previous works [40, 46], we fine-tune the last residual block of ResNet-50 [20] pre-trained on ImageNet-1K [9] for 20 epochs with the learning rate $10^{-3}$ while freezing the rest of the parameters. At inference time, all images are resized to 224×224. In Figure 1, we reported the performances of four OOD test datasets respectively. It can be seen that our method reaches state-of-the-art results on average across four OOD datasets.

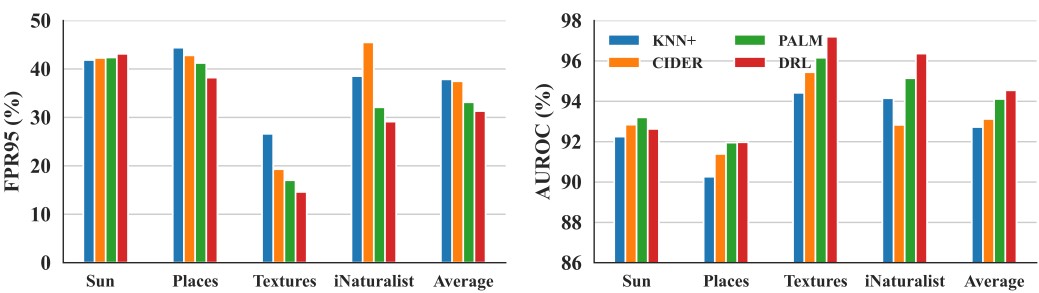

Figure 1: OOD detection results on the ImageNet-100 benchmark with ResNet-50.

## 5.3  Ablation study

In this section, we conduct an ablation study to validate our motivation and design. In Figure 2a, we compare the effect between $\ell_2$-normalzied and unnormalized class prototypes. While $\ell_2$-normalzied class prototypes enable DRL to avoid estimating the normalization constant, we can observe that unnormalized class prototypes come with a significant performance increase in detecting OOD samples. We suspect that unnormalized class prototypes come with a more reliable assumption that all classes are allowed to have a different level of concentration from each other. Besides, recalling that we have rearranged sequential sampling to optimize the ELBO of DRL in a consistent manner, Figure 2b empirically examines the effectiveness of this practice by demonstrating that the OOD detection performance considerably drops without the rearranging of sequential sampling.

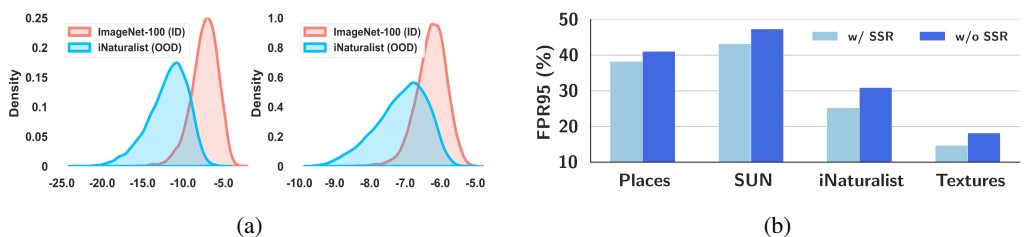

Figure 2: Ablation Study on the proposed DRL: (a) distribution of feature embeddings learned with unnormalized (left) and $\ell_2$-normalzied (right) class prototypes; (b) the benefits of the proposed sequential sampling rearrange to OOD detection performance.

## 6  Conclusion

In this paper, we propose a novel learning framework DRL that mitigates the gap between pre-trained discriminative models and density-based post-hoc OOD detection. Methodologically, DRL focuses on answering an important yet under-explored question of whether it is possible to deterministically shape the ID feature distribution while pre-training a discriminative model. Theoretically, DRL is formulated as an Expectation-Maximization algorithm, where we design a bounded ELBO and rearrange the sequential sampling for consistent optimization. Empirically, DRL achieves consistently strong performance of OOD detection compared to competitive baselines on multiple benchmarks, which implies the superiority of our proposed DRL. We hope our work can inspire future research on shaping ID feature space for density-based post-hoc OOD detection.

# 7 Acknowledgement

This work was supported in part by the NSFC / Research Grants Council (RGC) Joint Research Scheme under the grant N_HKBU214/21, the General Research Fund of RGC under the grants 12201321, 12202622, 12201323, and the RGC Senior Research Fellow Scheme under the grant: SRFS2324-2S02.

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

# A   Approximating vMF normalization constant

To approach the vMF probability density function for the random $d$-dimensional unit vector and concentration parameter $\epsilon$, one needs to get the normalization constant by calculating $\epsilon^{d/2-1} / \left[ (2\pi)^{d/2} I_{d/2-1}(\epsilon) \right]$ where $I_p(\cdot)$ denotes the modified Bessel function of the first kind and order $p$. However, $I_p(\cdot)$ is a complicated function. To cast off this dilemma, as explained by Eq. (9) in the main paper, we propose to approximate $I_p(\cdot)$ by resorting to an asymptotic expansion. We compare the values of $\log \hat{I}_p(\cdot)$ obtained by our proposed approximation with the values of $\log I_p(\cdot)$ obtained by the scipy implementation[2], where we define an approximation error $\delta(\epsilon)$ as follows:

$$\delta(\epsilon) = \log \hat{I}_p(\epsilon) - \log I_p(\epsilon) \tag{16}$$

It can be seen from Figure 3 that the approximation error is small compared to the actual function values when $p = 127$. This implies that the latent feature dimension $d = 256$ is sufficiently large to get a good approximation of the normalization constant.

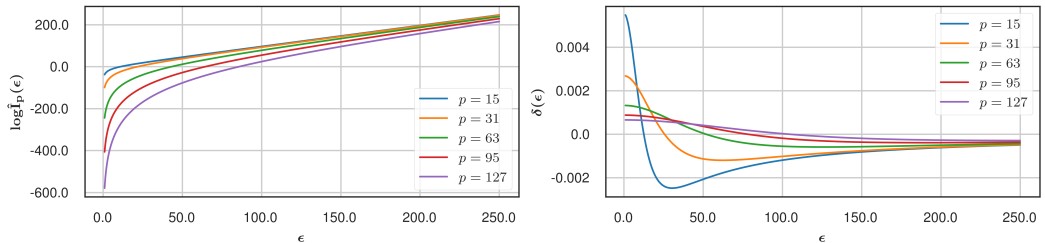

Figure 3: Left: The actual value of our proposed approximation of $\log \hat{I}_p(\epsilon)$; Right: The approximation error $\delta(\epsilon)$ in computing $I_p(\cdot)$ between our proposed and the scipy implementation.

# B   Training stability on CIFAR-100

To verify that our method consistently provides strong performance, we train with 5 independent seeds for CIFAR-100 and report the average and standard deviation of FPR and AUROC in Table 5.

Table 5: Ablation on stability. OOD detection performance of our method on CIFAR- 100. Results are averaged over 5 independent runs.

| Method | SVHN | | Places365 | | LSUN | | iSUN | | Texture | |
|--------|------|------|-----------|------|------|------|------|------|---------|------|
| | FPR95↓ | AUROC↑ | FPR95↓ | AUROC↑ | FPR95↓ | AUROC↑ | FPR95↓ | AUROC↑ | FPR95↓ | AUROC↑ |
| Mean | 10.15 | 98.07 | 66.64 | 81.55 | 6.97 | 98.63 | 22.57 | 96.33 | 21.97 | 96.09 |
| Std. | 1.52 | 0.11 | 3.06 | 2.41 | 2.28 | 1.46 | 3.25 | 0.87 | 2.24 | 1.94 |

# C   Limitations

This paper only explores one type of realization of the non-negative density function. It will be exciting to explore more possibilities for the realization and parameterization.

# D   Broader impacts

Our project aims to improve the reliability and safety of modern machine learning models. Our study can lead to direct benefits and societal impacts, particularly for safety-critical applications such as autonomous driving. Our study does not involve any human subjects or violation of legal compliance. We do not anticipate any potentially harmful consequences to our work. Through our study, we hope to raise stronger research and societal awareness towards the problem of out-of-distribution detection in real-world settings.

---

[2]https://docs.scipy.org/doc/scipy/reference/generated/scipy.special.iv.html

