# OpenReview forum: "Learning to Shape In-distribution Feature Space for Out-of-distribution Detection"
_NeurIPS.cc/2024/Conference — NeurIPS 2024 poster_

### Official Review · Reviewer_WmsK · 2024-06-22

**Soundness:** 4
**Presentation:** 3
**Contribution:** 3
**Rating:** 7
**Confidence:** 3

**Summary:**

This paper looks to perform OOD detection via distributional representation learning rather than assume some pre-specified distributional form for the ID data. Their motivations are that there can exist inconsistencies between the assumptions on the ID distribution from prior work and the actual unknown ground truth distribution.

They consider learning a feature space to approximate a mixture distribution over the ID data. The authors propose an online approach to approximate the normalization constant over the mixture distribution.

The authors also formulate this approach via a provably convergent EM algorithm to improve training stability. Specifically, they use latent variables as in-distribution classes, using Bayesian inference to derive the posterior distribution over latents given observed data (E-step). Then, the authors maximize the ELBO (which they also prove is bounded, as during each iteration of the algorithm, this value increases).

The authors empirically demonstrate the improvement of this approach over alternatives on average across multiple datasets when training on CIFAR10 and CIFAR100. They further show strong performance in hard OOD detection settings, where the goal is to detect data from CIFAR10 (and other datasets) when training on CIFAR100.

**Strengths:**

The authors show strong empirical results in average, compared to the considered baselines.

Their approach also circumvents the need for pre-specified distributional forms for the ID data, which (as the authors note) can be an issue in practice.

**Weaknesses:**

No apparent weaknesses.

**Questions:**

How quickly does this method converge? Does the EM algorithm require a comparable number of steps or computational complexity to the other considered approaches?

**Limitations:**

Yes, limitations are adequately addressed.

---

> ### Author Rebuttal · Authors · 2024-08-06
>
> We deeply appreciate the reviewer's dedicated time and effort in evaluating our work. In response to the insightful comments provided, we have provided detailed responses to each point raised, hoping that our responses can adequately resolve your concerns. Please find our responses below.
>
> **Q.1.** How quickly does this method converge?
>
> **A.1.** Thanks for your insightful idea. As can be seen from the loss trajectory shown in the uploaded PDF file, training with the same epochs as prior works [a,b] is sufficient for our method to converge. We will highlight the nature of our method following your valuable comments.
>
> [a] Learning with Mixture of Prototypes for Out-of-Distribution Detection, ICLR 2024
>
> [b] How to Exploit Hyperspherical Embeddings for Out-of-Distribution Detection, ICLR 2023

---

> > ### Comment · Reviewer_WmsK · 2024-08-12
> > **Reviewer Response**
> >
> > Thank you for the clarifications in the rebuttal! I'll maintain my score as an accept.

---

> > > ### Author Response · Authors · 2024-08-13
> > > **Thanks for your feedback**
> > >
> > > Dear Reviewer #WmsK,
> > >
> > > We're grateful for your quick feedback and deeply appreciate your consideration in keeping the score at 7 Accept.
> > >
> > > We remain open and ready to discuss any more questions or suggestions you might have. Your constructive comments have significantly improved the quality of our work.
> > >
> > > Best regards and thanks,
> > >
> > > Authors of Paper #8706

---

### Official Review · Reviewer_5Lwp · 2024-07-12

**Soundness:** 4
**Presentation:** 3
**Contribution:** 3
**Rating:** 7
**Confidence:** 4

**Summary:**

The paper provides a novel approach called DRL to help optimize post-hoc OOD detection methods by explicitly shaping the ID space during pre-training. In particular, DRL is defined through an Expectation-Maximization algorithm with alongside a structured mini-batch setting. The resulting DRL is shown to have strong empirical performance across multiple OOD detection methods and benchmarks. Additionally, the authors also provide a rigorous theoretical setup for justifying the DRL optimization schema.

**Strengths:**

Strengths:
- The paper provides a novel method for dealing with the problem of ID space shaping for OOD detection which is often overlooked in OOD detection.
- DRL shows strong empirical performance across multiple high-resolution benchmarks and traditional CIFAR benchmarks.
- The paper also provides a rigorous set of theoretical analyses to help better understand the underlying DRL method.

**Weaknesses:**

Weakness:
- A small concern of the reviewer is the formatting and structure of the paper. In particular, given the dense nature of the work, the reviewer would highly encourage some additional context to help introduce the reviewer to the DRL method.

**Questions:**

The reviewer has some confusion regarding the necessity of the sequential sampling alterations that the authors noted. In particular, the reviewer would like to get some more clarification regarding the inconsistency issues noted in Section 3.4.

**Limitations:**

The authors have adequately addressed limitations to the methodology as well as any negative societal impact. Additionally, the reviewer does not foresee any potential negative impact resulting from this work.

---

> ### Author Rebuttal · Authors · 2024-08-06
>
> We sincerely appreciate your dedicated time and effort in reviewing our work. In response to the valuable comments provided, we have provided detailed responses to each point raised, hoping that our responses can adequately address your concerns. Please find our responses below.
>
> **Q.1.** More clarification regarding the inconsistency issues
>
> **A.1.** We kindly note that the inconsistency stems from that, under the batch-based training scheme, the model used in the E-step of the $m$-th iteration is trained without the data used in the M-step of the same iteration. In particular, the prediction $p_{\boldsymbol{\theta}}(k|\mathbf{x})$ is only updated once per training epoch while the discriminative model $f_{\boldsymbol{\theta}}$ keeps updated throughout mini-batches (iterations) in the training.

---

### Official Review · Reviewer_7foW · 2024-07-14

**Soundness:** 2
**Presentation:** 3
**Contribution:** 2
**Rating:** 4
**Confidence:** 4

**Summary:**

The paper proposes an in-distribution (ID) modeling approach, termed distributional representation learning (DRL), which enhances the convergence of ID latent feature learning. The authors include theoretical proof to corroborate the proposed approach. They have conducted a few experiments on standard OOD benchmarks and explored different OOD cases.

**Strengths:**

1. The theoretical contribution enhances the idea of representation learning in the latent space. This theoretical proof helps explain related previous works that have proposed several ID modeling techniques.
2. The proposed method includes practical implementation techniques in the methods corresponding to the proposed theory.
3. The presentation is generally clear and easy-to-understand to readers.

**Weaknesses:**

1. Although the proposed data distribution modeling method is claimed to mitigate the distribution assumptions in the previous works, the model convergence still relies on prior works' assumptions such as vMF. Even though the techniques with underlying assumptions do not directly influence the geometry of the latent space, the proposed method is still not viable without these techniques. Other optimization strategies without the assumptions might be worth considering.
2. The improvements in the standard benchmarks are not significant and convincing. A few previous works' results are not consistent with the number in the papers. For example, using CIDER on the LSUN dataset obtains 30.24 FPR while 16.16 FPR was reported in the original paper for the CIFAR-100 ID data experiment.  The average FPR for Imagenet-100 is reported to be 25.9 in the original paper and GitHub repository which is much lower than the proposed method with 30.31 FPR in Table 3.
3. The ablation study and analysis of the proposed method can be improved. Figure 1(a) compares the effects of l2-normalization which has been widely known in most previous feature-based OOD detection methods. For Figure 1(b) observing the difference of sequence sampling, additional convergence analysis should be provided, such as loss trajectory or convergence time.
4. As described in line 189, the ELBO convergence issue might be a concern. Without sufficient evidence, it would be hard to justify the proposed method can completely avoid this kind of issue. As the proposed method is still a sampling strategy, the uncertainty of occasional unconverge situation might still occur.

**Questions:**

1. To perform online approximation, I would like to know if the batch size is an important parameter.
2. The description of $g_\theta$ in equation 7 (line 138) seems missing.
3. In Table 4, the proposed methods can hardly be comparable to the PALM in the ImageNet-Resize OOD detection results. Are there any reasons?
4. In Table 5, unsupervised settings seem to yield more failure cases, such as SVHN and LSUN datasets for the proposed method compared to PALM. Can the authors explain a bit about the results?
5. Why the authors did not include CIDER in Table 5?

**Limitations:**

The scope of OOD detection is widespread but the authors only consider simple and standard datasets. The experiments might introduce limitations to the study.

---

> ### Author Rebuttal · Authors · 2024-08-06
>
> We sincerely appreciate your time and effort in reviewing our work. Below are detailed responses to your valuable comments.
>
> **Q.1.** Although the proposed method is claimed to mitigate the assumptions in previous works, the model convergence still relies on prior works' assumptions (vMF). Even though the assumptions do not influence the geometry of the latent space, the proposed method is still not viable without these techniques.
>
> **A.1.** We apologize for the misunderstanding. We will add the following explanation to highlight our motivation.
>
> - Previous work assumes $p_{\theta}(x)$ (denoted as p_x in the left) to be the vMF distribution without theoretical guarantee. The mismatch between the assumed and learned distributions will degrade detection performance.
>
> -  Our method, with the derived ELBO (Eq. 10), enjoys the theoretical guarantee that the learned p_x conforms to vMF distribution. Here, introducing vMF is to compute the normalization constant in a closed form.
>
> - Our method can shape the learned p_x to fit various distributions, i.e., it is viable without the help of vMF. For example, if we, following [a], define $h(z,k)$ to be a Bregman divergence [b], the resulting p_x turns to be an exponential family distribution, which is left as our future work.
>
> [a] ConjNorm: Tractable Density Estimation for Out-of-Distribution Detection
>
> [b] Clustering with Bregman Divergences
>
> **Q.2.** The improvements in the standard benchmarks are not significant and convincing.
>
> **A.2.** We kindly note that the reported results of most compared baselines in all tables of this paper originate from PALM [c]. We are sorry for not explicitly illustrating this to make the reviewer confused.
>
> - While the reimplemented CIDER in PALM performs worse than the original CIDER on LSUN with CIFAR-100 as ID data, it is advantageous for the former to have better-averaged results across 5 OOD datasets than the latter. This is because OOD data tends to be from various domains rather than a specific one.
>
> Following PLAM, the experiments on ImageNet-100 are conducted on **ResNet-50**. However, the CIDER results reported on the GitHub repository are based on **ResNet-34** and thus can not be used for comparison. Thus, we **never** undergrad existing works.
>
> [c] Learning with Mixture of Prototypes for Out-of-Distribution Detection
>
> **Q.3.** The ablation study and analysis can be improved. Additional convergence analysis should be provided.
>
> **A.3.** Thanks for your constructive advice. We have added the loss trajectory to the revision, as shown in the PDF file.
>
> **Q.4.** It would be hard to justify the proposed DRL can completely avoid this issue. As it is still a sampling strategy, the uncertainty of occasional unconverge situations might still occur
>
> **A.4.**  We will add the following explanations.
>
> + The proposed DRL is formulated into a provably convergent EM framework. The inconsistency issue that occurs in the batch-based training scheme originates from the nature of the EM framework rather than our methodological design: during batch-based training scheme,  $p_{\theta}(k|x)$ is only updated once per epoch while $f_{\theta}$ keeps updated throughout batches (iterations). The EM framework has been widely used in deep learning literature [a,b,c,d,e,f,g,h], where inconsistency is an open problem.
>
> +  Our key idea is to update the model in the current iteration and take data from the upcoming iterations into consideration. This motivates sequential sampling to have $B\_m=(B\_m^{pre},B\_m^{next})$, where $B^{next}\_{m-1} = B^{pre}\_{m}$. In this way, by optimizing classification objective over $B\_m^{next}$ encourage $p\_{\theta_{m}}(k|x)$ to be close to the ground truth, the estimated $q\_{m+1}(k|x) = p\_{\theta_{m}}(k|x)$ can be reliable for $\mathcal{B}\_m^{next}$.
>
> +  Our key contributions show that one can deterministically shape p_x to conform to the known distribution defined via Eq. (7) to avoid the unalignment between the learned and assumed distributions. We also admit that the inconsistency issue is a fundamental topic, but addressing it with a theoretical guarantee is still an open problem in the literature and out of the scope of our work.
>
> [a] Knowledge Condensation Distillation
>
> [b] MiCE: Mixture of Contrastive Experts for Unsupervised Image Clustering
>
> [c] Prototypical Contrastive Learning of Unsupervised Representations
>
> [d] Joint Unsupervised Learning of Deep Representations and Image Clusters
>
> [e] Deep Clustering for Unsupervised Learning of Visual Features
>
> [f] Learning representation for clustering via prototype scattering and positive sampling
>
> [g] Stable Cluster Discrimination for Deep Clustering
>
> [h] Unsupervised Visual Representation Learning by Online Constrained K-Means
>
> **Q.5.** To perform online approximation, I would like to know if the batch size is an important parameter
>
> **A.5.** According to the definition in Line 159, the online approximation in Eq. (9) does not leverage data statistics within batches, i.e., it is independent of the batch size.
>
> **Q.6.** The description in eq 7 (line 138) is missing
>
> **A.6.** Thanks for pointing out the problem caused by typos. We have fixed this mistake:
> $p_{\theta}(x|k)=\frac{h(z,k)}{\Phi(k)},\quad \Phi(k) = {\int_{z \in \mathcal{Z}} h(z, k) \, d z}$, where $\mathcal{Z}$ is the latent feature space.
>
> **Q.7.** In Tables 4 and 5, the proposed methods can hardly be comparable to PALM
>
> **A.7.** We suspect this is because PLAM introduces more than one prototype for each ID class to learn better features regarding discriminating the aforementioned OOD dataset from the ID dataset. We kindly note that our method achieves a better average performance than PALM.
>
> **Q.8.** Why is CIDER not in Table 5?
>
> **A.8.** Unlike PALM proposed for unsupervised and supervised settings, CIDER is proposed for supervised settings. Thus, it is unclear how to extend CIDER to the setting. Thus, we exclude CIDER from Table 5.

---

> ### Author Response · Authors · 2024-08-13
> **The window for discussion is closing.**
>
> Dear Reviewer #7foW,
>
> ### **The window for discussion is closing.**
>
> Thanks very much for your great efforts in reviewing and valuable comments. The discussion will end soon. At this final moment, we would sincerely appreciate it if you could check our responses and new results regarding your concerns.
>
> **1. _The proposed method is still not viable without existing techniques._**
>
> We apologize for the misunderstanding. We have provided explicit explanations highlighting that the assumption used by previous methods is the motivation for our method instead of the basis for our work.
>
> **2. _The improvements are not significant and convincing_**
>
> We apologize for the misunderstanding. We have provided detailed explanations for the experimental results, i.e., we used results reported in the literature and the different results from the difference in model architectures.
>
> **3. _Explanations for sampling strategy_**
>
> We have provided detailed explanations highlighting the motivation of the proposed sampling strategy.
>
> If you have any further concerns, we will respond instantly at this final moment. We would sincerely appreciate it if you could confirm whether there are unresolved concerns.
>
> Best regards and thanks,
>
> Authors of #8706

---

> > ### Comment · Area_Chair_7eVv · 2024-08-13
> >
> > Hello Reviewer 7foW,
> >
> > Please take a moment to read and acknowledge the author's response to your review.
> >
> > Thanks, AC

---

### Official Review · Reviewer_s5Ce · 2024-07-15

**Soundness:** 3
**Presentation:** 2
**Contribution:** 3
**Rating:** 5
**Confidence:** 5

**Summary:**

The paper introduces an innovative learning framework, Distributional Representation Learning (DRL), designed to bridge the gap between network pre-training and density-based scoring strategies. DRL is formulated as a provably convergent Expectation-Maximization algorithm. Key contributions include the introduction of unnormalized class prototypes, which enhance the flexibility of the mixture model, and an online approximation of normalization constants, enabling end-to-end training. This framework represents a significant advancement in the field, offering both theoretical and practical improvements over existing methods.

**Strengths:**

The theoretical framework presented formulates DRL as an Expectation-Maximization (EM) algorithm, offering a robust foundation for the learning process. Recognizing the difficulty in directly optimizing the log-likelihood function, the authors adeptly shift focus to the optimization of $\text{ELBO}(q, x; \theta)$. To address potential inconsistency issues in this optimization, they introduce a sequential sampling rearrangement technique, significantly enhancing OOD detection performance. Additionally, One of the key strengths of the paper lies in its innovative approach to handling normalization constants. Rather than imposing impractical constraints to make these constants
input-independent or known, the authors propose an online approximation method, enabling seamless end-to-end training. The approach substantially enhances out-of-distribution (OOD) detection performance on CIFAR-10 and CIFAR-100 datasets.

**Weaknesses:**

- **Contradiction in Assumptions:** While the paper notes that existing methods often impose strict distributional assumptions due to the lack of prior knowledge about the learned feature space, it still enforces the underlying feature space to conform to a pre-defined mixture distribution. This approach appears to contradict its initial motivation.
- **Influence of Total Number of Classes (k):** The paper does not address the impact of the total number of classes (k) on computation and memory requirements. Including a detailed analysis of this aspect would provide a clearer understanding of the model's scalability and resource demands.
- **Performance Comparison with Other Density-Based Methods:** The paper lacks a clear comparison of DRL's performance with other density-based OOD detection methods.
- **Theoretical Proof for Sequential Sampling:** The paper lacks sufficient theoretical proof for the rearrangement of sequential sampling to ensure consistent optimization. Providing a rigorous theoretical foundation would strengthen the claims made in this regard.
- **Discussion on Hyperparameter $\beta$:** The paper does not include a discussion on the hyperparameter $\beta$. Including a comprehensive analysis of this hyperparameter, its impact on the model, and guidelines for its selection would be beneficial.
- **Equation Reference Correction:** In line 120, the reference to equation (7) should be corrected to equation (5).
- **Equation Mistakes:** In lines 176 and 183, there are errors in the equations that need to be addressed and corrected for accuracy.
- **Lack of Description for Table 5:** In line 288, under the section "Unsupervised OoD Detection," there is an insufficient description of Table 5. A more detailed explanation is required to clarify the contents and significance of the table.
- **Symbol Description:** In line 138, there is no clear description of the symbols in the formula. Providing detailed definitions and explanations of all parameters involved is essential for comprehensibility and reproducibility.

**Questions:**

- The paper initially criticizes existing methods for imposing strict distributional assumptions due to limited prior knowledge of the learned feature space. However, it then introduces a pre-defined mixture distribution for the underlying feature space. Can you clarify how this approach aligns with the initial criticism and motivation of your method?
- How does DRL's performance compare with other density-based OOD detection methods? Can you provide competitive performance metrics and analysis to evaluate DRL's relative effectiveness?
- What is the impact of the total number of classification classes (k) on computation and memory requirements? A detailed analysis would help in understanding the scalability and resource demands of your model.

**Limitations:**

- The paper primarily explores a single realization of the non-negative density function and imposes a pre-defined mixture distribution on the underlying feature space, which seems to contradict its initial motivation.Further exploration of different realizations could provide a more robust evaluation of the proposed approach.

- The paper includes experiments for large-scale OOD detection primarily on the ImageNet-100 dataset. If extended to the larger ImageNet-1k dataset, there are concerns that inference could become significantly slower and memory usage substantially higher. Addressing these potential scalability issues would be important for practical applications on large-scale datasets.

- There are various writing and expression errors throughout the paper. Improving the clarity and precision of the writing would enhance the overall readability and comprehension of the work.

---

> ### Author Rebuttal · Authors · 2024-08-06
>
> We sincerely appreciate your time and effort in reviewing. We have provided detailed responses below, hoping your concerns can be adequately addressed.
>
> **Q.1.** Contradiction in Assumptions
>
> **A.1.** We apologize for the misunderstanding and will provide explicit explanations highlighting that our method aligns with the motivation.
> - As shown in section 2, existing feature-based and logit-based methods explicitly or implicitly assume $p_{\theta}(x)$ as the Gibbs-Boltzmann distribution and Gaussian mixture distribution, respectively. However, there is no theoretical guarantee to ensure these distributional assumptions hold under any discriminative models $f_{\theta}$ that are trained by minimizing the classification objective. This inconsistency would degrade OOD detection performance.
> - Given that $p_{\theta}(x)$ is typically unknown after a discriminative model $f_{\theta}$ is trained by minimizing the cross entropy, our method proposes to deterministically shape $p_{\theta}(x)$ to conform to the known distribution defined via Eq. (7). This avoids the above unalignment issue. Namely, this leads to our motivation: is it possible to deterministically shape the ID feature distribution while pre-training a discriminative model? In this way, the defined distribution can be safely used for OOD detection since the involved distributional assumption can hold after training where Eq. (10) is minimized.
>
> **Q.2.** Influence of class number (K)
>
> **A.2.** Thanks for your kind suggestions. Our method's memory and time complexity linearly scales with K so that our method has the same scalability nature as the CE-based training.
>
> - The parameters involved in our method are located in 1) backbone and 2) prototypes $\mu\_1,...,\mu\_K$. Since the backbone is orthogonal to our technical design, we omit the memory complexity of the backbone. Therefore, the memory complexity of our method is $O(K)$.
> - Computing Eq. (14) for each x requires to compute $p\_{\theta}(x|k)$ for each $k$. By omitting the time complexity of the backbone and dot product, the time complexity of our method is $O(K)$.
> - We conduct ImageNet-100 for large-scale OOD, aiming to keep consistent with prior works [a,b].
>
> [a] Learning with Mixture of Prototypes for Out-of-Distribution Detection
>
> [b] How to Exploit Hyperspherical Embeddings for Out-of-Distribution Detection
>
> **Q.3.** Comparison with Density-Based Methods
>
> **A.3.** While this paper has considered popular post-hoc density-based methods, including Energy, Maha, and CONJ, for comparison, as per your valuable advice, we compare to more baselines (Maxlogit [c] and GEM [d]) and report the averaged results across 5 OOD datasets used in this paper as follows:
> || FPR95 | AUROC|
> | :-| :-|:-|
> | [d]|47.38|91.03|
> | [c] |33.55|90.97|
> |Ours|11.58|97.83|
>
> We compare it to the training-based density-based method [e], where an extra flow model is introduced on top of the pre-trained model. Here, following [e], we evaluate our method on CIFAR-10 and report the averaged results across 6 OOD datasets (5 used in this paper+LSUN-Resize)
> | | FPR95 | AUROC|
> | :-| :- | :-|
> | [e]|16.26|97.19|
> | Ours|13.19| 97.46|
>
> [c] Scaling Out-of-Distribution Detection for Real-World Settings
>
> [d] Provable Guarantees for Understanding Out-of-distribution Detection
>
> [e] FlowCon: Out-of-Distribution Detection using Flow-Based Contrastive Learning
>
> **Q.4.** Justification of Sequential Sampling
>
> **A.4.**  We will add the following explanations.
> + The proposed DRL is formulated into a provably convergent EM framework. The inconsistency issue that occurs in the batch-based training scheme originates from the nature of the EM framework rather than our technical design: during batch-based training scheme,  $p_{\theta}(k|x)$ is only updated once per epoch while $f_{\theta}$ keeps updated throughout batches (iterations). The EM framework is widely used in deep learning literature  [f,g,h,i,j,k,l,m], where inconsistency is an open problem.
> +  Our key idea is to update the model in the current iteration and take data from the upcoming iterations into consideration. This motivates sequential sampling to have $B\_m=(B\_m^{pre},B\_m^{next})$, where $B^{next}\_{m-1} = B^{pre}\_{m}$. In this way, by optimizing classification objective over $B\_m^{next}$ encourage $p\_{\theta_{m}}(k|x)$ to be close to the ground truth, the estimated $q\_{m+1}(k|x) = p\_{\theta_{m}}(k|x)$ can be reliable for $\mathcal{B}\_m^{next}$.
>
> [f] Knowledge Condensation Distillation
>
> [g] MiCE: Mixture of Contrastive Experts for Unsupervised Image Clustering
>
> [h] Prototypical Contrastive Learning of Unsupervised Representations
>
> [i] Joint Unsupervised Learning of Deep Representations and Image Clusters
>
> [j] Deep Clustering for Unsupervised Learning of Visual Features
>
> [k] Learning representation for clustering via prototype scattering and positive sampling
>
> [l] Stable Cluster Discrimination for Deep Clustering
>
> [m] Unsupervised Visual Representation Learning by Online Constrained K-Means
>
> **Q.4.** Ablation on $\beta$
>
> **A.4.** As per your advice, we conduct the ablation study on $\beta$ (with CIAR-10 as ID dataset) and show in the following:
> |$\beta$ |0.08|0.1|0.2|0.4|
> | :-| :-| :-| :-| :-|
> |Averaged FPR95|12.19|11.85|11.58|12.64|
>
> **Q.5** Description for Table 5
>
> **A.5.** In Table 5, we extend our method to the unsupervised setting. Following DINO [n], we keep a momentum teacher to produce soft labels as the target. As with DINO, we use the centering operation to avoid collapse.
>
> [n] Emerging properties in self-supervised vision transformers
>
> **Q.6** Equation Mistakes
>
> **A.6.** Thanks for your comments. The revised paper has fixed the typos and mistakes you mentioned.
>
> **Q.7** Symbol Description in line 138
>
> **A.7.** Thanks for pointing out the typo. We have fixed this in our revised paper and show the correct definition of $\Phi(k)$ in Eq. (7)
> $\Phi(k) = \int_{\mathbf{z} \in \mathcal{Z}} h(\mathbf{z}, k) d\mathbf{z}$

---

> ### Author Response · Authors · 2024-08-13
> **The window for discussion is closing.**
>
> Dear Reviewer #s5Ce,
>
> ### **The window for discussion is closing.**
>
> Thanks very much for your great efforts in reviewing and valuable comments. The discussion will end soon. At this final moment, we would sincerely appreciate it if you could check our responses and new results regarding your concerns.
>
> **1. _Contradiction in assumptions_**
>
> We apologize for the misunderstanding. We have provided detailed explanations highlighting that the proposed method aligns with our motivation.
>
> **2. _Influence of class number_**
>
> We apologize for the misunderstanding. Our method's memory and time complexity linearly scales with K so that our method has the same scalability nature as the CE-based training.
>
> **3. _Justification of sequential sampling_**
>
> Following your valuable suggestion, we have provided explicit explanations highlighting the motivation for the widely adopted approach.
>
> **4. _Comparison with density-based methods_**
>
> Following your valuable suggestion, we have added experiments, results, and discussions, demonstrating the effectiveness of our method.
>
> **5. _Ablation_**
>
> Following your valuable suggestion, we conducted the ablation study on the hyper-parameter $\beta$ mentioned.
>
> If you have any further concerns, we will respond instantly at this final moment. We would sincerely appreciate it if you could confirm whether there are unresolved concerns.
>
> Best regards and thanks,
>
> Authors of #8706

---

> > ### Comment · Area_Chair_7eVv · 2024-08-13
> >
> > Hello Reviewer s5Ce,
> >
> > Please take a moment to read and acknowledge the author's response to your review.
> >
> > Thanks, AC

---

### Author Rebuttal · Authors · 2024-08-07

We visualize our loss curve in the uploaded PDF file.

---

### Decision · Program_Chairs · 2024-09-25

**Decision:**

Accept (poster)

**Comment:**

The paper introduces a new framework called Distributional Representation Learning (DRL) for OOD detection by explicitly enforcing an underlying pre-defined mixture distribution via expectation-maximization.

The reviewers appreciated the innovative approach to OOD detection via in-distribution modeling, the practicality of the proposed procedure along with its strong empirical performance, and the theoretical backing for the procedure. The main reviewer concerns were sufficiently addressed in the author response.